# Joint single-cell DNA accessibility and protein epitope profiling reveals environmental regulation of epigenomic heterogeneity

Xingqi Chen[1,7], Ulrike M. Litzenburger[1], Yuning Wei[1], Alicia N. Schep[1,2,3], Edward L. LaGory[4], Hani Choudhry[5], Amato J. Giaccia[4], William J. Greenleaf [1,2,3] & Howard Y. Chang [1,6]

Here we introduce Protein-indexed Assay of Transposase Accessible Chromatin with sequencing (Pi-ATAC) that combines single-cell chromatin and proteomic profiling. In conjunction with DNA transposition, the levels of multiple cell surface or intracellular protein epitopes are recorded by index flow cytometry and positions in arrayed microwells, and then subject to molecular barcoding for subsequent pooled analysis. Pi-ATAC simultaneously identifies the epigenomic and proteomic heterogeneity in individual cells. Pi-ATAC reveals a casual link between transcription factor abundance and DNA motif access, and deconvolute cell types and states in the tumor microenvironment in vivo. We identify a dominant role for hypoxia, marked by HIF1α protein, in the tumor microvenvironment for shaping the regulome in a subset of epithelial tumor cells.

[1] Center for Personal Dynamic Regulomes, Stanford University, Stanford, CA 94305, USA. [2] Dept of Genetics, Stanford University, Stanford, CA 94305, USA. [3] Department of Applied Physics, Stanford University, Stanford, CA 94305, USA. [4] Division of Radiation and Cancer Biology, Department of Radiation Oncology, Stanford University, Stanford, CA 94305, USA. [5] Department of Biochemistry, Cancer Metabolism and Epigenetic Unit, Faculty of Science, Cancer and Mutagenesis Unit, King Fahd Center for Medical Research, King Abdulaziz University, Jeddah 22252, Saudi Arabia. [6] Howard Hughes Medical Institute, Stanford University, Stanford, CA 94305, USA. [7] Present address: Department of Cell and Molecular Biology, Karolinska Institutet, 17177 Solna, Sweden. These authors contributed equally: Xingqi Chen, Ulrike M. Litzenburger, Yuning Wei. Correspondence and requests for materials should be addressed to U.M.L. (email: litzenbu@stanford.edu) or to H.Y.C. (email: howchang@stanford.edu)

Cell-to-cell variation is a universal feature that impacts normal development and human disease[1]. While recent advances in single-cell research have improved our ability to document cellular phenotypic variation[1], the fundamental mechanisms that generate variability from identical DNA sequences remain elusive. Uncovering the molecular mechanism behind cellular heterogeneity would be helpful in clinical diagnosis, understanding the basic mechanism of developmental disorders, molecular basis of drug resistance in cancer, and therapy of human diseases in the long term. In the last decades, studies revealed that chromatin structure is a main player regulating gene expression, and that it is tightly linked to heterogeneity in transcription and phenotype[2]. To fully understand the molecular mechanism determining cell-to-cell heterogeneity, it is essential to define the chromatin landscape in each individual cell.

Recent advances in single-cell chromatin technologies revealed the variation of chromatin organization across individual cells[3–5]. These technologies demonstrate that accessibility variance is associated with specific transcription factors (TFs) and provide new insight into cellular variation of the "regulome"[3]. In these approaches, cells are randomly selected for next-generation sequencing and the cellular variation is decoded using computational de-convolution. Thus, using available technologies, we only interpret the cellular variation and define subtypes indirect by clustering, dimensionality reduction such as principal component analysis method or projection onto a bulk scaffold. Therefore, until now, the cell-to-cell epigenetic variation cannot unambiguously be linked to the cellular phenotype or cell state. Staining of proteins for specific cell types and cell stages is helpful to indicate the cellular phenotype, for example, phosphorylated focal adhesion kinase for a migratory cell state[6] or Hypoxia Inducible Factor 1 alpha (HIF1α) staining for cells in a hypoxic environment. Although an extensive effort was put on increasing throughput of these single-cell technologies[2,4], the direct linkage of cellular phenotype to the chromatin variation of individual cells remains largely ignored.

Here, we describe a novel single-cell approach, protein-indexed single-cell assay of transposase accessible chromatin-seq (Pi-ATAC), in which we index and quantify protein expression using index fluorescence-activated cell sorting (FACS) and enumerate the accessible DNA elements of the same individual cell. The combination of protein and epigenetic profile allows us to directly link the cellular phenotype and environment to the chromatin variation of individual cells. We applied Pi-ATAC to primary, heterogeneous mouse breast tumors and characterized cell states of tumor-infiltrating immune cells, as well as tumor cells simultaneously. In addition, we link epigenetic variability of tumor cells to the hypoxic tumor microenvironment. The described method allows to unbiasedly combine single-cell ATAC-seq with traditional FACS and therefore would be relevant to wide range of biology groups.

## Results

### Development of Pi-ATAC method.
We were motivated to develop Pi-ATAC to provide two innovative advances for multiomics. First, Pi-ATAC enables intracellular protein analysis and DNA accessibility from the same individual cell. We and others had used conventional flow cytometry with cell surface markers to isolate different cell types[7,8]. In Pi-ATAC, we have developed a new method to crosslink cells and perform intracellular protein analysis (including in the nucleus) jointly with single-cell ATAC-seq. Thus, Pi-ATAC opens the door for >85%[9] of the proteome for single-cell multiomics.

Second, in Pi-ATAC, we accomplish the indexing of both protein epitope levels and DNA regulatory landscape. Prior application of flow cytometry to ATAC-seq involved gates, where many cells within a wide range of protein levels are lumped together. This is a far cry from Pi-ATAC, where the level of individual protein epitopes in each cell is precisely enumerated.

Pi-ATAC works on fixed cells or tissue, which then can be stored prior to tagmentation, allowing collection of rare cells and pooling across multiple experiments. As a result, investigators can prospectively focus their sequencing power on rare but interesting cells. In more detail, in Pi-ATAC cells or tissue are first fixed using paraformaldehyde (PFA), then gently dissociated and permeabilized (Methods), followed by antibody staining against protein epitopes of interest. As the cells are already fixed and permeabilized, intracellular as well as intranuclear staining are possible. Then, cells are transposed in bulk. The reaction is stopped by addition of EDTA, without any purification step. As single cells are sorted into individual wells containing the reverse crosslinking buffer (Methods), fluorescence intensities of antibodies against protein epitopes of interest are recorded and assigned to the position of sorted cells. After reverse crosslinking, libraries are prepared by barcoding PCR (Fig. 1a).

The reverse crosslinking buffer used in Pi-ATAC was specially developed to be compatible with the barcoding PCR step. In ATAC-see, tagmentation is also performed on fixed cells, together with either intranuclear or cytoplasmic protein staining[10]. However, DNA Taq polymerase is not compatible with sodium dodecyl sulfate (SDS) included in the traditional reverse crosslinking buffer[11]. Hence, we developed a new reverse crosslinking (Methods), in which the DNA Taq polymerase is not inhibited.

First, we confirmed the functionality of the new reverse crosslinking buffer followed directly by PCR without purification in bulk ATAC-seq libraries. Quality of the bulk ATAC-seq libraries prepared using the new reverse crosslinking either with or without DNA purification is comparable to the previous protocol of reverse crosslinking buffer with purification, including (i) enrichment of fragments at transcription start sites, (ii) fragment amount observed in open chromatin peaks identified, and (iii) coverage on both gene promoters and distal elements (Supplementary Figure 1a-e). Together, we conclude that the new reverse crosslinking buffer is compatible with the PCR reaction for library preparation in Pi-ATAC.

### Validation of Pi-ATAC workflow in single cells.
We next validated the workflow and accuracy of FACS based scATAC-seq. In order to evaluate the precision of FACS sorting, we used a mixture of two cell types of different sizes and species ("barnyard experiment"): GM12878 human lymphoblastoid cell line (LCL) and V6.5 mouse embryonic stem cells (ESCs). Of note, the size difference between these two cell types is not a concern in the FACS-based scATAC-seq method, whereas the C1 microfluidics system requires separation into two size-restricted integrated microfluidic chips. First, both mouse ESC (mESCs, $n = 144$) and human LCL ($n = 144$) were fixed with 1% formaldehyde and mixed in a 1:1 ratio. Then ATAC was performed in bulk. Next, we sorted single cells from the mixture of cells by FACS without regard to cell type, and collected individual cells in 96-well plates containing the new reverse crosslinking buffer. After reverse crosslinking, library preparation and sequencing (Methods), all fragments were aligned to mouse and human genomes. Indeed, we found that each individual well contained predominantly either human or mouse DNA sequences, using a 500 fragment cutoff and 96% species specificity (0 hybrid out of 288 cells, Fig. 1b, Methods), confirming the precision of our index FACS sorting.

To assess the single-cell Pi-ATAC library quality, we prepared libraries from 192 GM17878 cells and comprehensively compared

those with bulk GM12878 ATAC-seq datasets ($n = 4$). Single cells that produced low-quality data were excluded from downstream analysis using nuclear fragment amount and fraction of reads in accessible chromatin peaks, as previously described[12]. Of the 192 GM12878 cells, 168 (87.5%) passed filter (Supplementary Figure 2a, Methods). Aggregation of these 168 single cells shows comparable accessibility patterns as bulk ATAC-seq (Supplementary Figure 2b, Supplementary Figure 2c $R = 0.81$ for 77,855 peaks, $p < 0.00001$). We then measured TF variation scores in GM12878 Pi-ATAC data, using the computational tool Chrom-Var[12] (Methods). The most significant variable TF motif was Nuclear Factor Kappa B (NF-κB) family (Supplementary Figure 2d), which agrees with previous scATAC-seq data from the same cell line[3].

Next, we compared Pi-ATAC data with publically available scATAC-seq data[3]. GM12878 Pi-ATAC and GM12878 scATAC-seq data yielded similar profiles of genome-wide accessibility ($R = 0.8$ for 77,855 peaks, $p < 0.00001$ Supplementary Figure 2e) as well as overall TF motif variability ($R = 0.7$ for 384 motifs, $p < 0.00001$ Supplementary Figure 2f). To ensure that antibody staining of extra- or intracellular proteins does not interfere with the quality of

Pi-ATAC, 384 GM12878 cells were stained against the B-cell surface marker CD19 and intracellular phosphorylated NF-κB (Supplementary Figure 2g). The antibody staining did not substantially affect the quality of Pi-ATAC, with 298 cells (77.6%) passing filter (Supplementary Figure 2h). The stained GM12878 Pi-ATAC data share concordance with previously published GM12878 scATAC-seq data in accessibility of peaks ($R = 0.72$ for 77,855 peaks, $p < 0.00001$ Supplementary Figure 2i), in the genomic annotation of peak distribution (Supplementary Figure 2j) and in TF motif accessibility at NF-κB and Jun motifs (Supplementary Figure 2k). Aggregation of these 298 single cells shows comparable accessibility patterns as bulk ATAC-seq (Fig. 1c). We next compared our GM12878 Pi-ATAC data with available scATAC-seq data across multiple cell types[3]. Two-dimensional t-distributed stochastic neighbor embedding (t-SNE) projection based on TF motif accessibility revealed clustering of single cells largely according to cell type, demonstrating Pi-ATAC GM12878 clusters together with the published scATAC-seq GM12878 data and distinct from other cell types (Supplementary Figure 2l).

We next quantified the information content provided by single cells in Pi-ATAC. Sparse DNA accessibility data from an

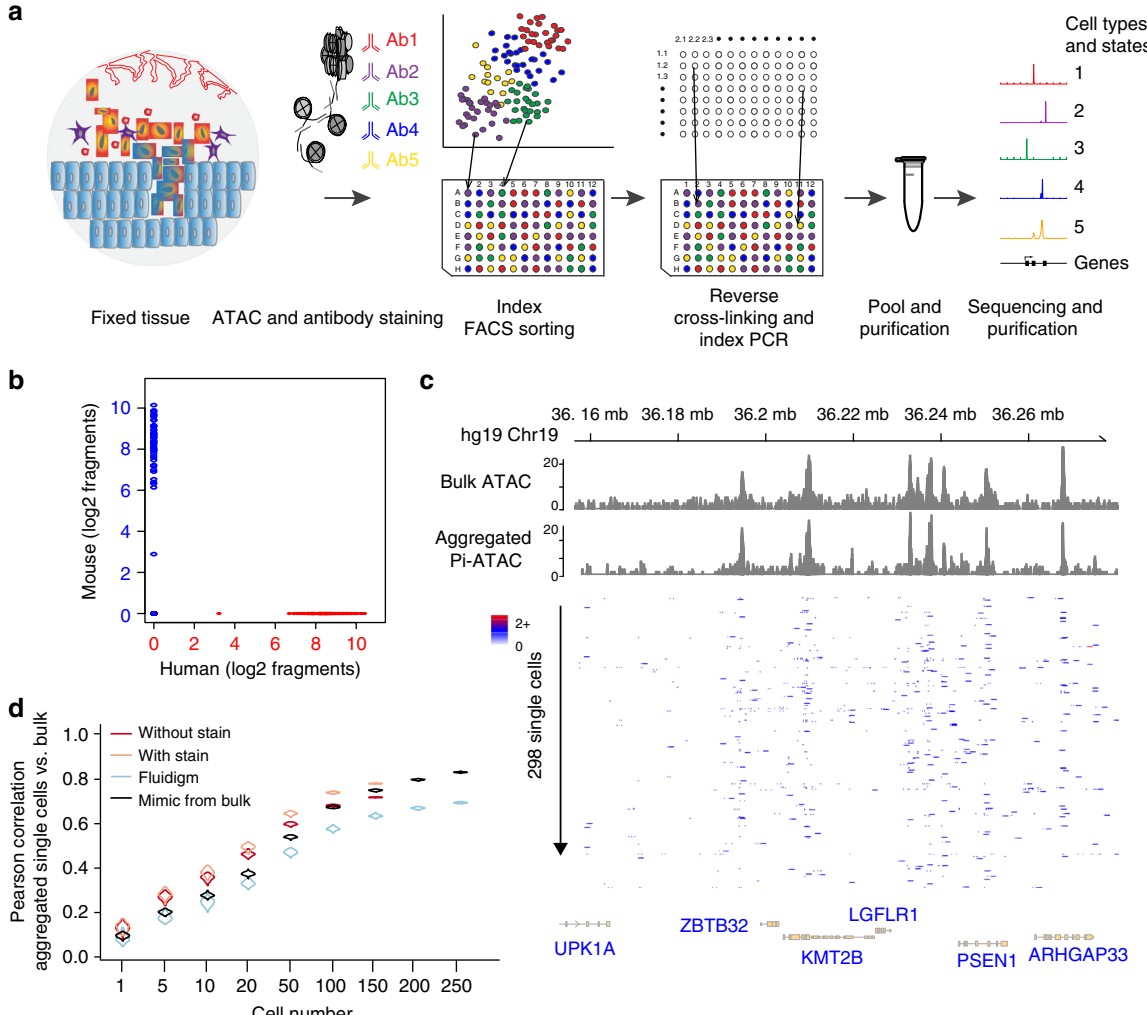

**Fig. 1** Principle of protein-indexed single-cell ATAC-seq (Pi-ATAC). **a** The workflow of Pi-ATAC. **b** The precision of single-cell FACS sorting demonstrated by aligned fragment comparison of mouse (mESC, blue) and human (GM12878, red) 1:1 cell mixture, using a 96% species and 500 fragment cutoff. **c** UCSC genome browser track comparison of aggregated 298 single-cell Pi-ATAC to bulk ATAC-seq (both GM12878). **d** Pearson correlation of fragment counts in ATAC-seq peaks in bulk GM12878 ATAC-seq data compared with aggregated single cells from indicated single-cell ATAC-seq approaches and mimic single cells from bulk (shown are distribution of 1000 times simulation on down sampling 500 fragments from each randomly selected single-cell or bulk sample)

individual cell suffice to infer TF motif activity, based on the summed accessibility of DNA elements containing the TF motif vs. background elements, as previously reported for scATAC-seq[12] (Supplementary Figure 2d, l). We find that data aggregation from 50 or more single cells by Pi-ATAC correlated significantly with bulk ATAC-seq on peak accessibility quantifications (Pearson correlation $R = 0.615$ for 77,857 peaks in 50 cells, $R = 0.694$ in 100 cells, $p < 0.0001$, false discovery rate

(FDR) < 0.01, Fig. 1d, Methods). This observation is controlled for cell-to-cell variance and sequencing depth artifacts, as determined by 1000 simulation runs of subsampling equalized contribution of 500 fragments in peaks per single-cell from a randomly selected 50 cells subset. The same analysis produced similar results in protein epitope-stained Pi-ATAC data vs. published GM12878 scATAC-seq data[3]. Similar results are obtained by down sampling to 500 fragments in peaks from

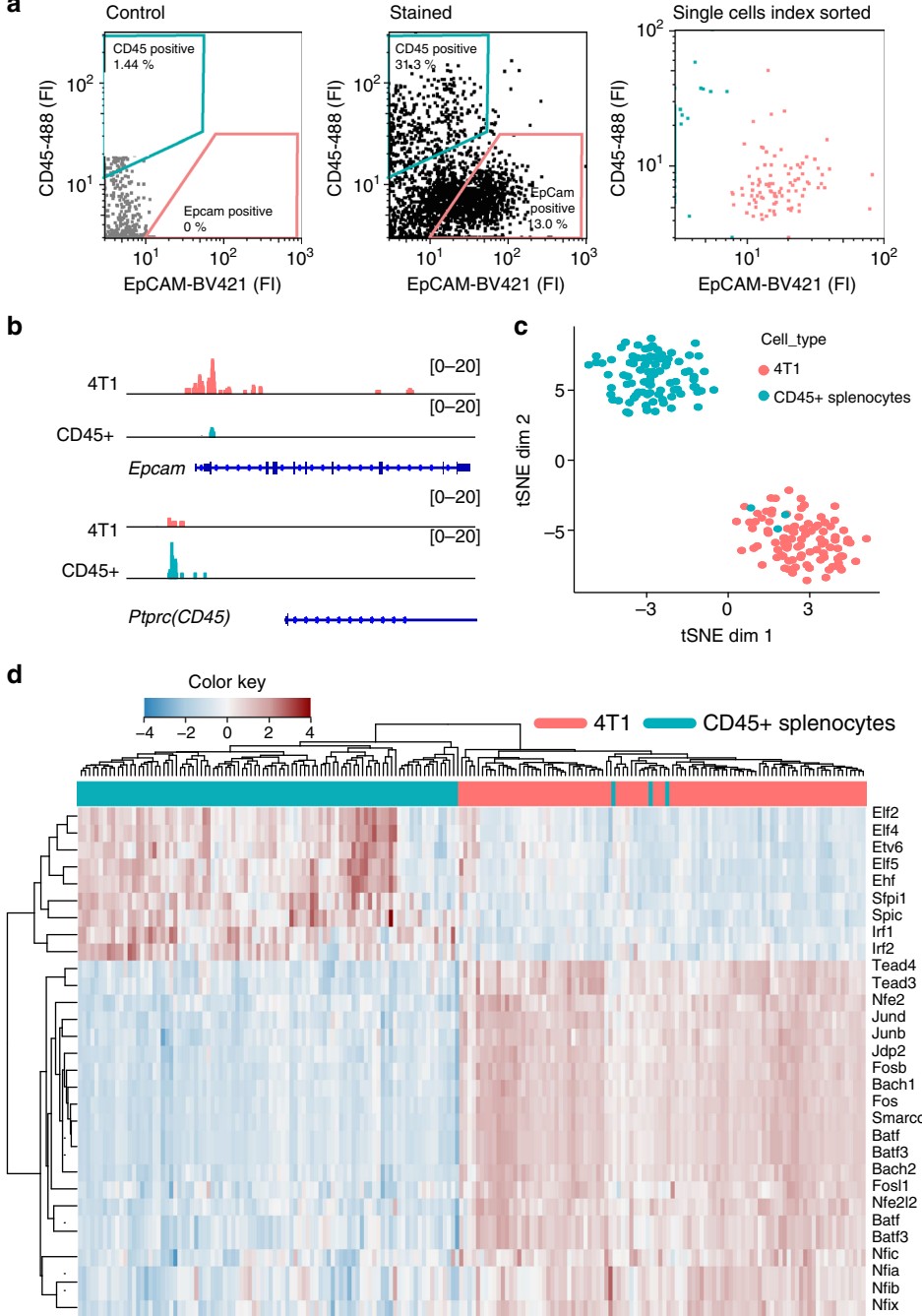

**Fig. 2** Pi-ATAC precisely dissects the cell types based on epigenetic profile. **a** FACS sorting gating strategy and data of EpCAM+ and CD45+ cells from mixture of 4T1 cells and mouse splenocytes; the histogram of protein staining is presented in Supplementary Figure 4a. **b** Genome browser tracks of aggregated Pi-ATAC 4T1 (EpCAM+) ($n = 95$) and splenocytes (CD45+) ($n = 95$) at the *Epcam* and *Cd45* loci. **c** t-SNE projection generated from TF deviation *z*-scores showing the projection of chromatin accessibility from Pi-ATAC 4T1 (EpCAM+) ($n = 95$) and splenocytes (CD45+) ($n = 95$). **d** Heatmap of unsupervised hierarchical clustering of the 30 top variable TF deviations from 4T1 (EpCAM+) ($n = 95$) and splenocytes (CD45+) ($n = 95$) Pi-ATAC. Each column represents a cell and each row a motif. The staining cluster information from FACS was assigned to each individual cell (top color bar)

bulk ATAC-seq data to mimic single-cell data ("mimic from bulk" in Fig. 1d, Methods).

Together, these results indicate that Pi-ATAC captures cell–cell variation in active regulatory DNA, as well as previous published scATAC-seq protocols while providing protein epitope measurements. With appropriate cell lineage and cell state characterization, Pi-ATAC data from appropriate single cells can be aggregated to reproduce a "pseudo-bulk" genomic DNA accessibility profile.

**Pi-ATAC probes TF protein abundance vs. DNA motif access.** The K562 chronic myeloid leukemia (CML) cell line is a heterogeneous cell mixture containing both precursor and differentiated cells[13]. Several key TFs such as GATA1, GATA2, or TAL1 regulate hematopoietic precursor status[14,15]. Along with others, we demonstrated that the self-renewal and multipotency of K562 cells is correlated with high expression level of GATA2[16,17]. It is unclear whether GATA2 is a pioneer factor in this context, i.e., creating the first DNA access at its cognate motif, or follows other TFs by accessing DNA elements occupied by other TFs. Therefore, we applied Pi-ATAC to K562 cells using GATA2 staining to measure the epigenetic status and heterogeneity of the precursor cells within the K562 population (Methods). To focus only on the nuclear presence of GATA2, we isolated the K562 nuclei to exclude cytoplasmic GATA2, as well as mitochondria contamination.

As expected, GATA2 staining intensity in K562 nuclei is variable, ranging from 1 to 8400 relative fluorescence units , clearly clustering into three groups, which we categorized as high, medium, and low (Supplementary Figure 3a). Of the 288 nuclei sorted, 223 (77.4%) passed the filter (Supplementary Figure 3b). In line with previous scATAC-seq results[3], our Pi-ATAC results display highest variability in the motif accessibility of the GATA and Jun/FOS family (Supplementary Figure 3c, d, Methods). Surprisingly, we did not find a significant correlation between accessibility of GATA2 motif itself with GATA2 staining (Supplementary Figure 3e), but we observed significant positive correlations of GATA2 protein level with accessibility of other TF motifs, including TAL1::TCF3 ($R = 0.21$), MTF1 ($R = 0.19$), CEBPA ($R = 0.12$), GSC ($R = 0.15$), SRF($R = 0.16$), and DUX($R = 0.26$, all with $p < 0.05$ and FDR $< 5\%$), which are involved in maintaining hematopoietic cells in a de-differentiated cell state[14,15,18–20] and determining hematopoietic cell fate in multi-potential progenitor cells[21] (Supplementary Figure 3e). TAL1, TCF3, and MTF1 have been reported to be potential binding partners in larger protein complexes with GATA2[22,23]. At the same time, we also observed that the accessibility of TFAP motif is significantly anti-correlated with GATA2 stain ($R = -0.2$, $p < 0.05$) (Supplementary Figure 3e). TFAP was previously reported to drive hematopoietic differentiation[24], the opposite role of GATA2 in CML cells. Based on direct quantification of nuclear GATA2 protein and chromatin access from the same cell, our results suggest that GATA2 cooperates with other TFs such as TAL1 and MTF1 to promote CML cell self-renewal in K562 cells.

Our findings suggest the feasibility to use Pi-ATAC to directly probe TF mechanisms linking protein abundance, and DNA accessibility.

**Simultaneous Pi-ATAC of tumor cells and immune cells.** We next applied Pi-ATAC to primary tumor samples. Tumor tissue is comprised of different cell types, including tumor, stromal, and tumor-infiltrating immune cells, each with heterogeneous cell sizes and cell surface markers. Therefore, we first tested Pi-ATAC on an artificial mixture of mouse breast tumor cell line 4T1 and

mouse splenocytes for (i) the sort efficiency of cells with comparable size differences as occurring in the primary tumors, (ii) lineage-specific antibodies, such as the mutually exclusive expression of EpCAM in the breast cancer cells and CD45 in splenocytes and tumor-infiltrating immune cells. As expected, the mixture of breast cancer cells and splenocytes are clearly distinguished and sorted using EpCAM and CD45 antibodies (Fig. 2a, Supplementary Figure 4a). We sorted 192 cells and performed Pi-ATAC, of which 190 (99.0%) cells passed the filter (Supplementary Figure 4b). Importantly, aggregated single cells from EpCAM+ 4T1 and CD45+ splenocytes show cell type-specific open chromatin at *Epcam* and *Cd45* loci, respectively (Fig. 2b), confirming the specificity of the antibodies and sorting. Within the top 30 out of 278 significant variable TF motifs, we observed two clusters of chromatin accessibility profiles specific for 4T1 and splenocytes, respectively, Spic and Irf motifs in immune cells and Nfi family motifs in 4T1 tumor cells (Fig. 2d). Addition of the protein-staining information confirms the chromatin accessibility clustering and also reveals that three splenocytes (1.5%) cluster with 4T1 cells (Fig. 2c, d). Together, these results show the specificity and precision of Pi-ATAC approach to deconvolute cell communities.

**Pi-ATAC of breast tumor ecosystem.** Cancer heterogeneity is a major driver of tumor evolution, progression, and emergence of drug resistance. Although cancer heterogeneity can be assessed with recent advances in single-cell technologies, the focus lies mainly on cell-to-cell transcriptional or mutational differences. The heterogeneity of epigenetic information within a tumor, particularly in solid tumors, is largely unexplored.

As a case study, we applied Pi-ATAC to the fast growing and highly invasive *MMTV-PyMT* genetically engineered mouse breast tumor model[25] to simultaneously dissect EpCAM+ tumor cells and tumor-infiltrating immune cells (CD45+) (Fig. 3a, Methods). We stained a single-cell suspension of the dissected tumor with anti-EpCAM and anti-CD45 as above and performed Pi-ATAC on 384 cells (Supplementary Figure 5a). We did not exclude any tumor regions or cells, obtaining an unbiased view of the composition of the tumor. Of all stained cells 4% were immune cells (CD45+), 28% were epithelial tumor cells (EpCAM+) (Supplementary Figure 5a). The quality of Pi-ATAC in these primary tumor cells isolated from a solid tumor was equivalent to Pi-ATAC performed of any cell line with 369 (96.1%) cells passing the filter (Supplementary Figure 5b). Aggregated single cells from EpCAM+ and CD45+ cells show cell type-specific open chromatin at *Epcam* and *Cd45* loci, respectively (Supplementary Figure 5c).

Surprisingly, although the distinction of the protein staining is pronounced, the t-SNE projection of EpCAM+ cells and CD45+ cells of bias-corrected deviations for motifs did not form two distinct clusters (Fig. 3b). However, the protein information aids visualization of a polarity of the distribution of the EpCAM+ cells and immune cells and reveals mixing of the two cell types. The previous observed clear separation of 4T1 and mouse splenocyte epigenetic profiles excludes the possibility of antibody cross-contamination.

We next calculated the variability of TF motifs across all cells, resulting in 84 significant variable motifs ($p < 0.05$ after Benjamini–Hochberg (BH) correction on multiple tests, Supplementary Data 1). Unsupervised hierarchical clustering of the 84 TF motifs across 369 cells revealed three modules of motif accessibilities (28 motifs in m1, 14 in m2, and 42 in m3) and seven subgroups of single cells (s1–s7 with cluster size range from 27 to 68 cells) (Fig. 3c, d, Supplementary Figure 5d, i). TF motifs of m1 consisting of immune

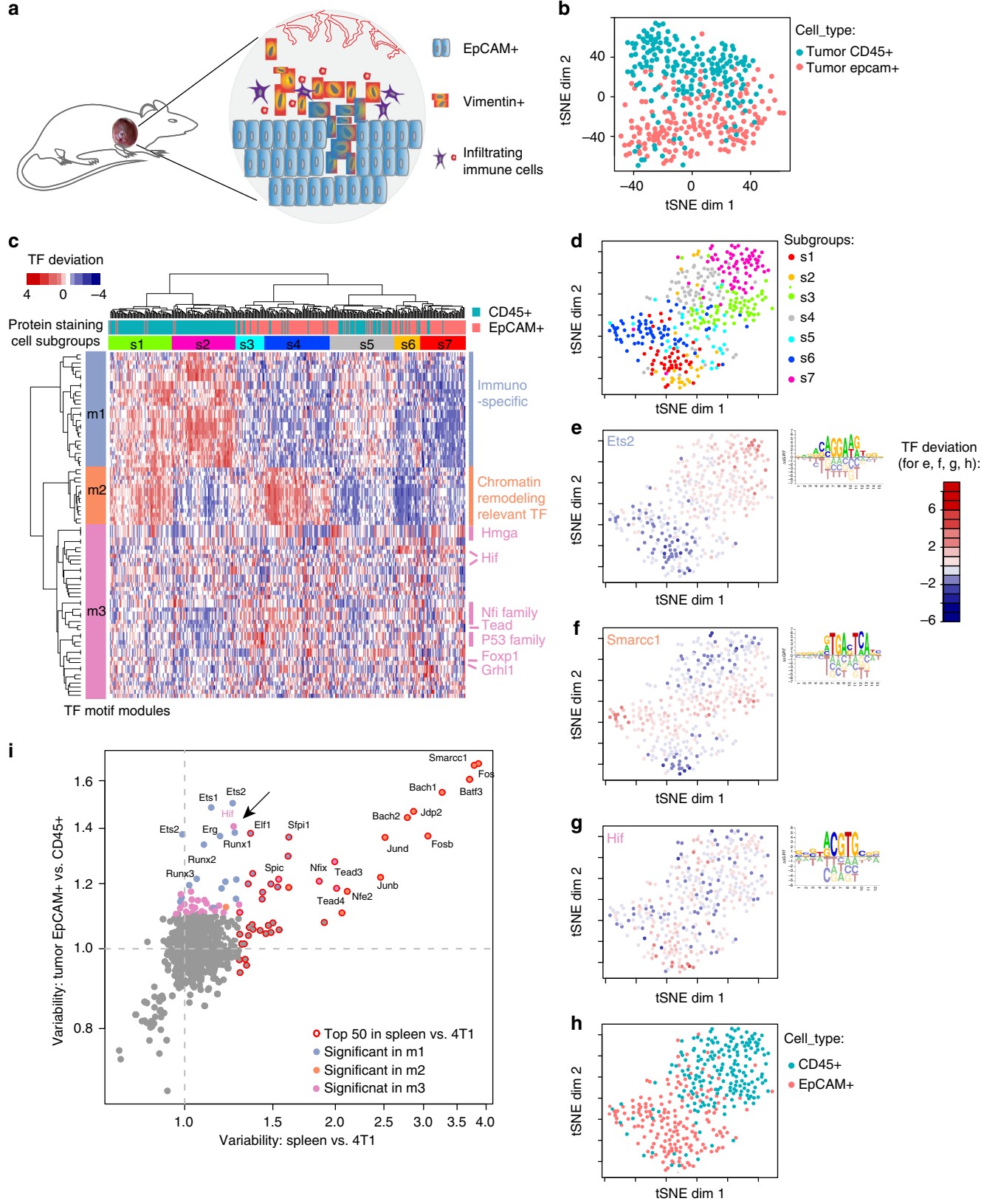

cell-specific motif families (Spic, Runx, Ets, etc.) show uniform accessibility among cells in s1 and s2 (Fig. 3c, e). M2 motifs, including chromatin remodeling relevant TF motifs such as Smarcc1, Jun, and Fos family are consistently accessible in cells of s1 and s4 (Fig. 3c, f). M3 comprises TF motif families Hmga,

Hif, Nfi, Tead, p53, Grhl1, and Fox. We observed heterogeneity across all cellular subgroups for TF motif accessibility in m3, however, highest mean motif accessibility in s3, s4, and s6 (Fig. 3c, g, Supplementary Figure 6). Collectively, the seven different cellular subgroups might reflect the heterogeneity of

**Fig. 3** Pi-ATAC dissects EpCAM+ tumor cells and tumor-infiltrating immune cells from the same mouse breast tumor. **a** Schematic illustrating the different cell types in the mouse breast tumor. **b** t-SNE projection of TF deviation $z$-scores of Pi-ATAC EpCAM+ ($n = 177$) and CD45+ ($n = 192$) cells from the same mouse breast tumor. **c** Unsupervised hierarchical clustering of the TF deviation $z$-scores of all 84 significant variable TFs ($p < 0.05$ after Benjamini–Hochberg (BH) correction on multiple tests) across EpCAM+ ($n = 177$) and CD45+ ($n = 192$) cells from the same mouse breast tumor. Each column represents one cell and each row a transcription factor motif. Motif modules (m1–3) and cell subgroups (s1–7) are marked with distinguished colors. In addition, the staining cluster information from FACS was assigned to each individual cell (top color bar). **d** t-SNE projection of TF deviations of 84 significant variable TF motifs of EpCAM+ and CD45+ cells isolated from a tumor, color coded by cellular subgroup information; **e–g** color coded by the accessibility of the TF motif with most significant variability in each module (Fig. 3d): Ets2 in m3 (**e**), Smarcc1 in m2 (**f**) and Hif in m1 (**g**). Motifs are based on PWM from CisBP database. Red is highly accessible, blue is low accessible; **h** color coded by immune-phenotype. **i** Scatter plot of TF motif variability calculated by ChromVAR across 4T1–splenocyte mixture to TF variability calculated by ChromVAR across EpCAM+ – CD45+ primary tumor cells. Colors indicate TF motif modules as in (**c**). Arrow points to Hif

the composition of EpCAM+ tumor cells and CD45+ tumor-infiltrating immune cells.

Next, we aligned the FACS protein-staining information to the seven cellular subgroups (Fig. 3c, h). As expected, s1 and s2 are significantly enriched with CD45+ cells (92.4% of 66 cells in s1 and 94.1% of 68 cells in s2, chi-squared test $p < 0.0001$ for others, after BH correction of multiple tests, Supplementary Data 2). The majority (102 out of 177 (57.6%)) of EpCAM+ cells fall into s3, s4, and s7 (76.7% of 30 cells in s3, 88.2% of 68 cells in s4, and 90.7% of 43 cells in s7, enriched with EpCAM+ cells, chi-squared test $p < 0.0001$ for others, after BH correction of multiple tests, Supplementary Data 2). The remaining 75 of 177 EpCAM+ cells are in s5 and s6 mixed together with CD45+ cells. EpCAM+ subgroups are characterized by accessibility of TF motifs in m3, a module characterized by heterogeneous accessibility in various TF motifs such as p53-, Tead-, Nfi-, and Hif-families. This heterogeneous TF motif accessibility in m3 prevents clear categorization of some cells, but the protein information guides the classification into immune and tumor cells. Interestingly, m2 motifs associated with chromatin remodeling TFs, such as Smarcc and Fos, show specific accessible in subgroups of both EPCAM+ (s1) and CD45+ (s4) cells (Fig. 3f), suggesting variable regulation on different subgroups within each cell type. Our result shows that Pi-ATAC dissects chromatin profile of different cell types simultaneously from the primary tumor, and staining information is necessary to unambiguously define the cell type from such complex system.

We also performed an unbiased analysis of sequence features associated with variation in chromatin accessibility across tumor and immune cells. We searched for enrichment of short nucleotide sequences of a specific length $k$ in accessible regions using ChromVAR. This k-mer analysis allows discovery of novel motifs and is not restricted by any database entry. Because most TF have core motifs between 5 and 8 bp, we use 6-, 7-, and 8 k-mer analysis to find enrichment of motifs of 8 or less base pairs (Supplementary Data 3-5). The top enriched ($p$-value $< 0.05$ after BH correction of multiple tests) k-mer represents RUNX motif family, followed by TEAD, Jun-Fos, and NFI families (Fig. 3c, Methods). As all of these k-mers are already found using annotated TF motifs in CisBP database, we decided to proceed using CisPB results for downstream TF motif analysis.

Assuming the artificial mixture of immune and tumor cells defines TF motif accessibility characteristic for these cell types, we compared TF motif variability of 4T1 with splenocytes, and of CD45+ tumor-infiltrating immune cells to EpCAM+ primary tumor cells (Fig. 3i). Of the 84 significant variable TF motifs across the primary cells, 49 are shared between the two datasets. Of note, although the top 10 most variable motifs from the tumor immune cell mixture experiment are all m2 motifs shared with the primary cells, most of the heterogeneous TF motif

accessibility in m1 and m3 was not observed as significantly variable in the cell line mixture. Thus, we next asked which TF motifs are the main contributors to the heterogeneity in m3 and m1 assuming that contributing TFs are unique features of in vivo tumors. Motifs along the $y$ axes stand out as having additional variability of accessibility in tumor cells and infiltrating immune cells compared with the artificial mixture (Fig. 3i). Of these motifs, the most variable ones are defining m1 of the hierarchical clustering (Fig. 3i, lavender dots). These results show that motifs accessible in m1 are unique to the tumor-infiltrating immune cells, because of their absence in the splenocytes. Although m1 motifs are accessible specifically in CD45+ cells, m3 accessibility is biased towards EpCAM+ cells (Supplementary Figure 6). Interestingly, the predominant TF motif specific to primary tumor cells enriched in m3 is Hif (Fig. 3i, pink dots). We found the Hif motif to be the most significant unique variable motif of the heterogeneous m3 module (Fig. 3g). HIF protein stability is tightly linked to the hypoxic environment[26], thus we hypothesize that the hypoxic microenvironment in a solid tumor may modulate the chromatin accessibility and heterogeneity observed in the primary tumor.

**Hypoxia drives single-cell epigenetic variability in tumor microenvironment.** To directly demonstrate that the hypoxic microenvironment influences the tumor epigenetic landscape in vivo, we applied Pi-ATAC to EpCAM and HIF1α-stained cells in the *MMTV-PyMT* breast tumor model. We isolated 956 EpCAM+ cells and observed 762 negative, 139 low, and 55 high HIF1α protein-positive cells from the tumor (Fig. 4a). Among these, 956 cells analyzed by Pi-ATAC, 839 passed the filter (Supplementary Figure 7a).

Focusing first on the accessibility profiles, unsupervised hierarchical clustering revealed two motif clusters and 20 cell state subgroups (Supplementary Figure 8a–c). The first motif cluster is dominated by high accessibility of motifs of the E26 transformation-specific (ETS) family, whereas the m2 is not showing clusters of accessible motifs. The subdivision of all EpCAM+ cells into 20 subgroups suggests high epigenetic heterogeneity within the tumor cells. Interestingly, HIF1α staining does not correlate with any specific cell cluster, demonstrating that the protein information provides an orthogonal measurement to the epigenetic profile.

The variability of TF motif accessibility profiles across the three HIF-staining groups is substantially different (Fig. 4b, Supplementary Data 6). This difference is not due to the variable sample size of the HIF-staining groups as demonstrated by down sampling simulations (Supplementary Figure 7b, Supplementary Data 6). The variability of TF motif accessibility is the highest among cells not experiencing hypoxia, which are also the majority of cells in the population (Fig. 4b, Supplementary Figure 7b). Among the strongest variable motifs in the negative HIF1α group

are Fos, Smarcc1, and Batf, which were observed to be accessible in a subgroup of EpCAM+ cells in the previous experiment. The variability of low and high HIF1α is dominated by p53 TF family motifs. Hypoxia is known to increase p53-protein levels via

several mechanisms, which may be reflected in TF motif activity[27]. Additional TF motifs that are highly variable in hypoxic cells include Grainyhead and Snail, TFs involved in epithelial–mesenchymal transition and known mediator of breast

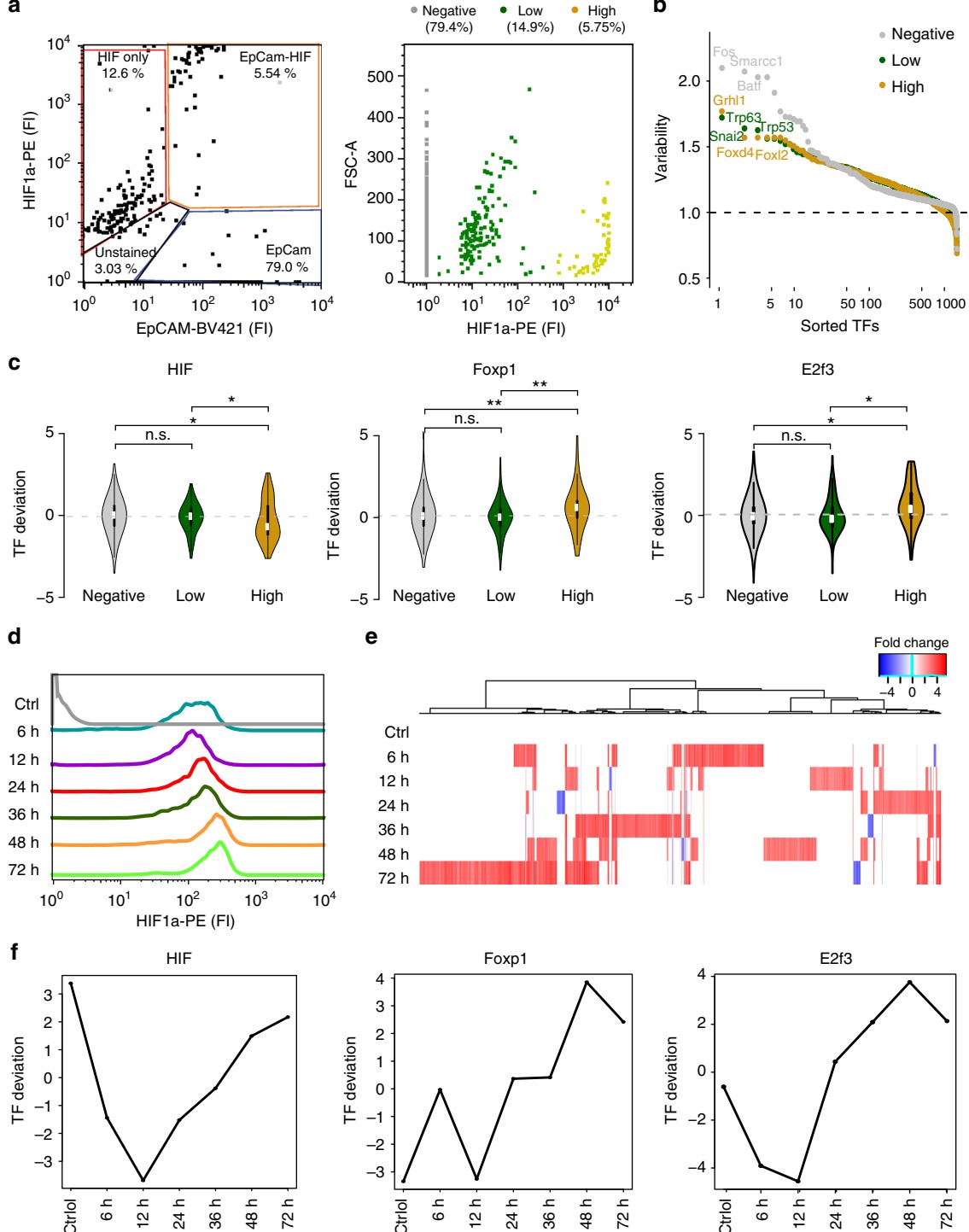

**Fig. 4** Epigenetic variability is modulated by the hypoxic microenvironment. **a** FACS sorted EpCAM+ and HIF1α+ double-positive cells from the same mouse breast tumor; three groups of HIF1α staining were assigned, HIF1α negative ($n = 762$), HIF1α low ($n = 139$), HIF1α high ($n = 55$) (right plot). **b** Ranking of transcription factor motif variability of different HIF1α-staining groups, HIF1α negative ($n = 702$), HIF1α low ($n = 95$), HIF1α high ($n = 42$). **c** Examples of TF motif deviations, which significantly change in HIF1α high group compared with the others. In each comparison, HIF1α negative ($n = 702$), HIF1α low ($n = 95$), HIF1α high ($n = 42$). Significance of Wilcoxon test shown with n.s. (not significant), *$p < 0.05$ and **$p < 0.01$. **d** Flow cytometry analysis of HIF1α protein abundance in 4T1 breast cancer cells cultured in 5% oxygen (Ctrl, gray) or 1% oxygen for indicated time points. **e** Chromatin accessibility changes of hypoxic conditions compared with control. Red indicates higher, blue lower accessibility. **f** TF deviation examples of motifs whose accessibility increases in hypoxia over time in 4T1 breast cancer cells (TF deviation was calculated using ChromVAR)

cancer metastasis[28,29] and might reflect the evading mechanism observed in hypoxic cells[27].

Fox and E2f3 were previously reported to be regulating transcription of HIF target genes[30,31]. Indeed, the tumor Pi-ATAC data demonstrate a significant increase of DNA accessibility of motifs for Fox family and E2f3 in the cells with high HIF1α protein (Wilcoxon test $p < 0.05$, Fig. 4c) in a more detailed comparison of TF motif accessibility across the three groups. At the same time, we observed TF motifs of Pit1 and Atoh8 lost significant DNA accessibility (Wilcoxon test $p < 0.05$) in cells with high HIF1α protein (Supplementary Figure 9). Atoh8 expression was reported to be reduced in hypoxic conditions[32]. Pit1 is an important developmental factor[33] associated with cancer but not previously associated to be regulated by the HIF network.

Surprisingly, we found on average decreased accessibility at HIF motifs in HIF1α high cells compared with negative and low (Fig. 4c, Wilcoxon test $p < 0.05$). However, we also observe a bimodal distribution of HIF motif accessibility within this HIF1α high group (Fig. 4c, $p < 0.05$ for Hartigans' dip statistic). This feature is not due to variable sample size across the three staining groups, as shown by down sampling simulation in negative and low HIF1α clusters to the same number of cells as found in the HIF1α high group ($n = 42$) (Supplementary Figure 10, Supplementary Data 7). This bimodality shows heterogeneity in HIF motif accessibility, which might reflect distance to the hypoxic center or time of hypoxia exposure.

By organizing single cells based on HIF1α protein levels, we now can interpret a subset of DNA accessibility dynamics that likely reflect the hypoxic microenvironment modulating cooperating TFs acting on target gene promoters and enhancers.

To confirm if the DNA accessibilities of tumor cells, particularly HIF, Fox, E2f3, Atoh, Pit1 motifs, are indeed modulated by hypoxia and no other tumor microenvironmental stimuli, we used an in vitro cell line culture system to simulate the in vivo hypoxia microenvironment. We incubated the mouse breast tumor cell line 4T1 in 1% $O_2$ for various time points (6, 12, 24, 36, 48, and 72 h). To avoid the degradation of HIF proteins under normoxia, we fixed and collected cells in 1% $O_2$ followed by HIF1α protein staining and standard bulk ATAC-seq at the indicated time points. Increased HIF protein stability was then measured by flow cytometric analysis (Fig. 4d). We observe an increase in HIF1α staining, confirming that the 4T1 cells experienced bona fide hypoxia.

All bulk ATAC-seq libraries have good reproducibility at all the time points (Supplementary Figure 11a–c). We observe changes in the chromatin accessibility across time in low oxygen, with 635 at 6 h to over 1000 sites opening at 72 h compared with normoxic cells. Interestingly, only a small number of DNA elements (~20–70) lose DNA access during this treatment (Fig. 4e). These differential accessible chromatin sites were annotated to 755 significantly differential accessible TF motifs, using the CisBP database (Supplementary Figure 11d, Supplementary Data 8, Methods).

This reference time course allowed us to next focus on the dynamic changes of TF motifs, which we assume to be involved in hypoxic responses of tumor cells, because they are enriched in HIF1α high group in the Pi-ATAC tumor experiment (Fig. 4c). Indeed, we found that the chromatin accessibilities are affected at HIF, Fox, E2f3, Ptx1, and Atoh8 motifs during the different time points of hypoxia incubation (Fig. 4f, Supplementary Figure 12). Also in vitro HIF motif accessibility undergoes dynamic changes, showing both decrease and then increase at different time points of hypoxia treatment confirming the slight decrease and variability observed in vivo. Foxp1 and E2f3 motif gain DNA accessibility with increasing duration of hypoxia incubation (Fig. 4f). At the same time, accessibility of Pit1 and Atoh8 motifs

decrease with longer hypoxia incubation, both results confirming the accessibility differences observed in HIF1α high vs. negative group in vivo (Supplementary Figure 12).

In summary, Pi-ATAC is the first technique that allows directly linking the microenvironment to epigenetic heterogeneity in a tumor.

## Discussion

Here, we develop a novel single-cell technology, Pi-ATAC, which simultaneously measures protein epitopes and active DNA regulatory elements of the same individual cell. Pi-ATAC enables investigation of the basic molecular mechanism of cellular heterogeneity from multiple angles.

Pi-ATAC directly links the cellular phenotype or microenvironment to the epigenetic profile, because it uses antibody staining and index FACS sorting to combine protein abundance information with the epigenetic profile of a cell. Powerful applications are exemplified by our experiments quantifying the protein levels of TFs NF-kB and HIF1α, both regulated by intricate post-translational modification, and measuring their DNA occupancy at the same time. Furthermore, we demonstrate that Pi-ATAC is applicable to both fixed cells from cultured cells and solid tumors in vivo, without any cell size restrictions.

In Pi-ATAC, choice of antibodies allows to combine phenotypic characterization of cells with their epigenetic profile. Pi-ATAC can directly link protein abundance or modifications with chromatin landscape to address mechanisms of gene regulation. We demonstrate that the epigenetic profile does not necessarily have a one-to-one relationship with the "phenotype" indicated by protein stain, demonstrating that both together give a more complete picture of cell state than either one alone. The impact of this precision analysis is highlighted in our analysis of tumor hypoxia, where the intermediate level of HIF1 protein induction is linked to tumor suppression pathway.

The tumor microenvironment has been recognized as an integral component of malignancies in mammary and other carcinomas, contributing in confounding ways to tumor progression, metastasis, therapy resistance, and disease recurrence[34]. In the tumor microenvironment, immune cells change from an immune-active to an immune-suppressive state, stromal cells turn into tumor-promoting tissue. The altered phenotype of these cells is proposed to be mainly due to epigenetic deregulation of gene expression;[35] however, until now, a direct measurement linking the microenvironment with epigenetic changes does not exist. These epigenetic changes are highly dynamic, causing heterogeneity of cell states within a tumor, which contributes to drug resistance, and tumor recurrence[36]. By directly linking the environment with HIF1α protein accumulation and the chromatin profiles of single cells, we show that tumor hypoxia dominantly shapes the regulome profiles of both parenchymal tumor cells and infiltrating immune cells, becoming the dominant epigenomic signature for a minority of cells in the tumor. Moreover, different levels or durations of hypoxia can induce distinct TF activity states that expand tumor heterogeneity. The biological insight of this new technique will be of great value for single-cell research in future.

In summary, Pi-ATAC is easily scalable, fast and economical (~$0.15 per cell–$1.50 per cell, Methods). Pi-ATAC is high throughput and easily scalable; with sorting into 384-well plates and use of 96 × 90 adapter combinations to index 8640 single cells in one experiment. Importantly, Pi-ATAC can be performed without any specialized instrument or reagents, allowing ready access to the broad community. Pi-ATAC may be useful to address potential prognostic and therapeutic opportunities revealed by these insights in future studies.

## Methods

**Cell culture**. GM12878 were purchased from Coriell Institute and grown in RPMI-1640 (11875-093, Gibco), 2 mM L-glutamine (25030-081, Gibco), 15% fetal bovine serum (FBS, 07905, Gibco), and 1% Pen/Strep (15140-122, Gibco). 4T1 mouse breast cancer cells were purchased by ATCC (CRL-2539) and cultured in RPMI-1640 supplemented with 10% FBS (07905, Gibco) and 1% Pen/Strep (15140-122, Gibco). K562 human CML cells were purchased from ATCC (CCL-243) and cultured in Iscove's Modified Dulbecco's Media supplemented with 10% FBS (07905, Gibco) and 1% Pen/Strep. V6.5 mESCs (Novus Biologicals) were cultured in Dulbecco's modified Eagle's medium/F12 supplemented with N2, 10 ng/ml basic fibroblast growth factor (R&D Systems), and 1000 U/ml of Leukemia inhibitory factor (LIF). For the 4T1 in vitro hypoxia experiments, cell lines were cultured in a Ruskinn Invivo2 workstation maintained at 1% $O_2$ and 5% $CO_2$, and cells were collected with trypsin digestion in the Ruskinn Invivo2 workstation and fixed immediately after 6, 12, 24, 36, 48 and 72-h incubation.

**Mice**. MMTV-PyMT mice were ordered from the Jackson Laboratory and bred and housed in the Stanford University Research Animal Facility in accordance with the guidelines (APLAC #14046). The laboratory animal care program at Stanford is accredited by the Association for the Assessment and Accreditation of Laboratory Animal Care (AAALAC International).

**Splenocyte isolation**. A spleen was dissected from a MMTV-PyMT mouse, which did not have a tumor at the time of euthanization. Then, the spleen was sliced into small pieces. Single cells were isolated from these pieces by placing them onto a 70 μm cell strainer and mechanically pushing the cells with a plunger through the strainer. Red blood cells were permeabilized using ACK buffer (Thermo Fisher, A1049201). After washing cells twice with RPMI-1640 medium, cells were counted and used in the 4T1–splenocyte mixture Pi-ATAC experiment.

**Mouse breast tumor dissection and dissociation**. Tumor growth has been monitored and mice were sacrificed at 3 month of age, when tumors reached a 1 $cm^3$ size. The mouse was immediately perfused using 4% PFA, which was subsequently quenched with 0.125 M glycine. The tumor was dissected and first cut into small pieces on ice, followed by enzymatic digestion using 200 U Collagenase V in Hanks' Balanced Salt Solution (HBSS) + 0.1% FBS for 1 h at 37 °C rotating. After centrifugation cells were resuspended in PBS and this suspension was filtered through a 70 μm cell strainer to remove any remaining clumps. Cells can be stored in phosphate-buffered saline (PBS) for at least 1 week at 4 °C.

**Bulk ATAC-Seq**. Fixed ATAC-seq was performed as previously described[10]. GM12878 cells were fixed with 1% formaldehyde (Sigma, USA) for 10 min and quenched with 0.125 M glycine for 5 min at room temperature. After the fixation, cells were counted and 50,000 cells were used per ATAC-seq reaction. The transposition reaction follows the normal ATAC-seq protocol. After the transposition, a reverse crosslink solution (final concentration of 50 mM Tris-Cl, 1 mM EDTA, 1% SDS, 0.2 M NaCl, 5 ng/ml proteinase K) was added up to 200 μl. The mixture was incubated at 65 °C with 1200 rpm shaking in a heat block overnight, then purified with Qiagen Mini-purification kit and eluted in 10 μl Qiagen EB elution buffer. Sequencing libraries were prepared following the original ATAC-seq protocol[37]. The sequencing was performed on Illumina Hi-Seq at the Stanford Functional Genomics Facility.

**Bulk Pi-ATAC-seq without elution**. GM12878 cells were fixed with 1% formaldehyde (Sigma, USA) for 10 min and quenched with 0.125 M glycine for 5 min at room temperature. After the fixation, cells were counted and 50,000 cells were used per ATAC-seq reaction. For bulk Pi-ATAC of 4T1 cells, cells were first fixed in hypoxia, then counted and permeabilized, followed by transposition. The transposition reaction is the same as for standard ATAC-seq except with 0.05% Igepal CA-630 in the lysis buffer. After 30-min transposition, at 37 °C, the reaction was quenched using 40 mM EDTA, then cells were centrifuged and supernatant was discarded. In all, 20 μl of reverse crosslinking solution was added to the cell pellet (with final concentration of 50 mM Tris-HCl pH 8.0, 0.5% Tween 20, 0.5% Igepal CA-630, 5 ng/ml proteinase K). Reverse crosslinking was performed 65 °C overnight and inactivated by a 10-min incubation at 80 °C the next day. The 25 μl PCR master mix (NEB, M0541S) and 5 μl of two unique primer combinations were directly added to the reverse crosslinking mixture and PCR was performed as previously described[37]. The PCR product was purified with Qiagen MinElute purification kit and eluted in 20 μl Qiagen EB elution buffer. The 75 × 2 paired-end sequencing was performed on Illumina HiSeq4000 at the Stanford Functional Genomics Facility.

**Pi-ATAC of mixture of GM12878 and mES cells**. In all, 50,000 GM12878 and 50,000 mESCs were mixed 1:1. The mixture was fixed with 1% formaldehyde (Sigma, USA) for 10 min and quenched with 0.125 M glycine for 5 min at room temperature. Then, cells were permeabilized using standard ATAC lysis buffer (10 mM Tris ph7.5, 10 mM NaCl, 3 mM $MgCl_2$, fresh 0.1% NP40) and immediately spun down. Cells were then transposed in bulk; for 100,000 cells 2× ATAC

reactions were calculated. After 30-min tagmentation, reaction was quenched by addition of 40 mM EDTA. Cells were centrifuged, supernatant discarded, and pellet resuspended in PBS for single-cell index sorting. In the analysis, all alignments below 500 fragments were disregarded.

**Pi-ATAC of mixture of 4T1 cells and splenocytes**. In total, 50,000 4T1 and 50,000 splenocytes were mixed 1:1. The mixture was fixed with 1% formaldehyde (Sigma, USA) for 10 min and quenched with 0.125 M glycine for 5 min at room temperature. Then, cells were permeabilized using standard ATAC lysis buffer (10 mM Tris ph7.5, 10 mM NaCl, 3 mM $MgCl_2$, fresh 0.1% NP40) and immediately spun down. Cells were then transposed in bulk; for 100,000 cells 2× ATAC reactions were calculated. After 30-min tagmentation, reaction was quenched by addition of 40 mM EDTA. After that, cells were stained at room temperature with anti-mouse EpCAM-BV421 (BD Horizon, #563214) CD45-Alexa488 (BioLegend #103122, clone 30-F11) for 30 min. Cells were then washed twice, the final pellet resuspended in PBS for single-cell index sorting.

*Pi-ATAC of GM12878 cells*. A detailed step-by-step Pi-ATAC protocol can be found at [https://www.protocols.io/private/F59D7D2F8FD5E57A20E039E9CF7A9785]. GM12878 cells were counted and 100,000 cells were fixed with 1% formaldehyde (Sigma, USA) for 10 min and quenched with 0.125 M glycine for 5 min at room temperature. Then, cells were permeabilized using standard ATAC lysis buffer (10 mM Tris ph7.5, 10 mM NaCl, 3 mM $MgCl_2$, fresh 0.1% NP40) and immediately spun down. For indicated experiments, 100,000 cells were then stained at room temperature with anti-human CD19-PE Clone H1B19 (BioLegend 302207) and rabbit anti-human Phospho-NF-kb p65 (Ser536, Cell Signaling) followed by donkey anti-rabbit 488 secondary antibody. After staining (30 min each antibody), cells were transposed in bulk; for 100,000 cells 2× ATAC reactions were calculated. After 30-min tagmentation, reaction was quenched by addition of 40 mM EDTA. Cells were centrifuged, supernatant discarded, and pellet resuspended in PBS for single-cell index sorting.

**Index sorting and library preparation for Pi-ATAC**. Single cells were index sorted into 96-well plates without any specific staining selection using the FACS AriaII (BD Biosciences). The plates were pre-filled with 20 μl reverse crosslinking buffer (see above) per well. For each experiment, index sort files (fcs) were exported using BD FACS Software and then further analyzed by FlowJo and R.

For reverse crosslinking, all 96-well plates were incubated overnight at 65 °C, then proteinase K was inactivated by 10-min incubation at 80 °C.

Single-cell libraries were prepared by adding 25 μl 2 × PCR Master Mix (NEBNext High fidelity, NEB) and 2.5 μl of 25 mM barcoding primer to each well. PCR cycling conditions: 72 °C for 5 min; 98 °C for 30 s; 20 cycles at 98 °C for 10 s, 63 °C for 30 s, and 72 °C for 1 min. After that, all wells were pooled, purified using the MinElute kit from Qiagen and eluted in 20 μl Qiagen EB elution buffer. The 75 × 2 paired-end sequencing was performed on Illumina HiSeq4000 at the Stanford Functional Genomics Facility. On average, we sequenced ~1000 single cells on one HiSeq 4000 lane, resulting in an average of ~0.3 millions reads per cell.

*Pi-ATAC of K562 cells*. First, we isolated the K562 nuclei by following the protocol as reported[38]. For the Pi-ATAC, 200,000 nuclei were transposed; for 100,000 nuclei 2× ATAC reactions were calculated. After 30-min tagmentation, reaction was quenched by addition of 40 mM EDTA. After that, cells were stained at room temperature with mouse anti-human GATA2 (Abnova, 1-102) 30 min at room temperature, followed by donkey anti-mouse 488 secondary antibody. Cells were then washed twice, the final pellet resuspended in PBS for single-cell index sorting.

*Pi-ATAC of tumor cells*. In all, 1×$10^6$ cells isolated from a PyMT mouse breast tumor (see above) were stained for 30 min at room temperature for rat anti-mouse EpCAM-BV421 (BD Horizon, #563214), HIF1α-PE (R&D, #IC1935P), CD45-Alexa488 (BioLegend #103122, clone 30-F11), followed by donkey anti-rabbit 488 secondary antibody. After staining, cells were transposed in bulk. Here, for 4000 final cells we tagmented 400,000 cells, meaning 8× ATAC reactions were calculated. After 30-min tagmentation, reaction was quenched by addition of 40 mM EDTA. Cells were centrifuged, supernatant discarded, and pellet resuspended in PBS for single-cell index sorting.

Single cells were index sorted into 96-well plates without any specific staining selection using the FACS AriaII (BD Biosciences). The plates were pre-filled with 20 μl reverse crosslinking buffer (see above) per well. For each experiment, index sort files (fcs) were exported using BD FACS Software and then further analyzed by FlowJo and R.

For reverse crosslinking all 96-well plates were incubated overnight at 65 °C, then proteinase K was inactivated by 10-min incubation at 80 °C.

Single-cell libraries were prepared by adding 25 μl 2 × PCR Master Mix (NEBNext High fidelity, NEB) and 5 μl of two unique primer combinations, the PCR cycling conditions: 72 °C for 5 min; 98 °C for 30 s; 20 cycles at 98 °C for 10 s, 63 °C for 30 s, and 72 °C for 1 min. After that, all wells were pooled, purified using the MinElute kit from Qiagen and eluted in 20 μl Qiagen EB elution buffer.

*Pi-ATAC library purification, quantification, and sequencing.* The pooled single-cell Pi-ATAC libraries were loaded on a 6% polyacrylamide gel electrophoresis, run at 160 V for approximately 10-min purification, stain the gel with SYBR Gold. One 0.5 ml tube was prepared for one sample, and a hole was made in the bottom of 0.5 ml tube with 20 gauge needle. The gel slice with range of above 150-bp DNA were cut and put in the 0.5 ml (with a hole in the bottom). Next, the 0.5 ml tube is put in the 2 ml tube and centrifuged with max speed for 3 min at room temperature. After removing the 0.5 ml tubes, 300 μl buffer (500 mM NaCl, 1 mM EDTA, 0.5% SDS) was added in the gel. The gel is incubated at 55 °C overnight with 1400 rpm shaking. Next day, the gel is centrifuged with max speed for 3 min at room temperature, and the supernatant is recovered for DNA purification with ChIP DNA Clean & Concentrator Kits (ZYMO RESEARCH, D5205). The recovered DNA is quantified with Agilent High Sensitivity DNA ChIP and sequenced on Illumina Hi-Seq at Stanford Functional Genomics Facility.

*Pi-ATAC cost per cell.* For Pi-ATAC, reagent costs is largely driven by the PCR master mix. Cost estimate per cell: Tn5 enzyme (Illumina, $0.00049 per cell, assuming 2.5 μl Tn5 per 50,000 cells), NEBNext® High-Fidelity 2X PCR Master Mix (NEB, M0541) and primers (~$1.30 per cell) and negligible costs for antibodies per cell (about $3 per protein per $10^6$ cells). In summary, it costs about $1.50 per cell with 50 μl PCR reaction system. In addition, we had succeeded with low-scale volume PCR reaction system at 5 μl per well of a 96-well plate, where the cost is ~$0.15 per cell.

*Bulk Pi-ATAC data preprocessing.* Paired-end reads were trimmed for Illumina adapter sequences and transposase sequences using custom-written script and mapped to hg19 or mm9, respectively, using Bowtie2[39] v2.1.0 with parameters —"very sensitive". Reads were subsequently filtered for alignment quality of > Q30 and were required to be properly paired. Duplicate reads were removed with Picard (http://picard.sourceforge.net) v1.79. Reads mapping to the mitochondria were removed and not considered. The uniquely mapped reads are merged before peak calling if bulk samples origin from the same cell/tissue/condition. Peak calling was performed by MACS2[40] narrow peak mode with parameters –q 0.01 –nomodel –shift 0. Peaks were filtered for following categories: (1) mitochondrial fragments inserted in nuclear genomic sequences, by Nuclear MiTochondrial Sequences (NumtS) [http://hgdownload.cse.ucsc.edu/goldenPath/hg19/database/ or http://hgdownload.cse.ucsc.edu/goldenPath/mm9/database/] and (2) the consensus excludable ENCODE blacklist genomic regions [http://mitra.stanford.edu/kundaje/akundaje/release/blacklists/hg19-human/hg19-blacklist.xls or http://mitra.stanford.edu/kundaje/akundaje/release/blacklists/mm9-mouse/mm9-blacklist.bed.gz].

To process comparison among bulk samples from multiple cell/tissue/conditions, peak sets from all conditions were merged by bedtools merge to a consensus peak list. Number of raw reads mapped to each peak in each sample was quantified using multicov in Beltools. Peak raw counts were normalized using DESeq2.

*Pi-ATAC data processing.* All single-cell Pi-ATAC data were preprocessed with custom-written script as previously described[3] and bulk preprocessed. Briefly, pair-ended reads were trimmed with adapter sequences, were mapped to hg19 or mm9 with Bowtie2 v2.1.0 with parameter-X2000. Uniquely mapped reads were filtered for low-quality reads, duplicates, and reads mapped to mitochondrial genome. Peak summit calling was processed with aggregated single cells of one cell/tissue condition, by MACS2 narrow peak mode with parameters –q 0.05 –nomodel –shift 0 –call summits. We annotated peaks as the summit-centered 500 base pair long genomic region. The top 50,000 peaks with least *q*-value reported by MACS2 were included for downstream analysis in most datasets; whereas top 30,000 peaks were included for K562 dataset. Single-cell libraries were further filtered with requirement of at least 500 fragments mapped to genome and at least 0.5 times the median proportion of fragments in peaks reached by aggregated single cells of the same cell/tissue condition (ChromVAR)[12].

**Pi-ATAC data information content analysis**. The quantification of fragments mapped to each peak for the genomic DNA accessibility was compared between bulk GM12878 Pi-ATAC samples and single-cell Pi-ATAC. Peak quantification of bulk data was calculated as mean value of the four samples. Peak quantification of Pi-ATAC was calculated for single cell, as well as for single cells aggregation in group size 5, 10, 20, 50, 100, and 150. To adjust cell–cell variance, 1000 times of simulation were processed to randomly select cells from the 192 GM cells into the group. To adjust sequencing depth artifact, 500 of fragments passing the final filter were randomly subsampled from the bam file for each cell in each time of simulation. The same process was conducted for public scATAC-seq GM12878 data[3]. We also generated 50 "mimic" single-cell data through down sampling 500 fragments passing the final filter directly from each of the four bulk samples. Pearson correlation coefficient with significance was calculated between peak quantification of bulk and single-cell data from each approach.

*TF deviation and variability analysis.* Single-cell Pi-ATAC data processing and calculation of TF deviation was performed using the R package ChromVAR as reported[12]. TF motifs were derived from the JASPAR dataset for human data,

whereas Homer and CisBP databases were used for mouse data, with R package motifmatchr. Briefly, for each TF, the accessibility in each cell was calculated by subtracting the number of Pi-ATAC reads in peaks covering the corresponding motif in the cell, and was normalized by the total TF accessibility in the cell. The accessibility deviation value for the TF in each cell is subtracted by the mean accessibility calculated for sets of background peaks with similar accessibility and GC content to obtain a bias correction, and further divided by standard deviation of the deviations calculated for the background peak sets.

TF deviation for the hypoxic time course bulk ATAC-seq was calculated using the R package ChromVAR.

The *Z*-score of deviations for each TF is used for visualization in both t-SNE projection plots and heatmaps of unsupervised hierarchical clustering based on correlation metric, by R packages Rtsne and pheatmap.

The variability of the TF motif across single cells in the sample set was determined by the standard deviation of the TF deviations across the cells. The metrics is close to 1 if the motif is not significantly more variable compared with the background peak list of that motif. To remove artifact of cell amount difference among the three groups of mouse breast tumor assembled by HIF1α staining, 100 times of simulation were processed through down sampling 42 cells from HIF1α low and HIF1α median group to re-calculate TF variabilities.

*k-mer analysis.* We performed an unbiased analysis of nucleotide sequence features necessary for chromatin accessibility variation by searching for an enrichment of short nucleotide sequences of a specific length *k* in accessible regions using ChromVAR. This k-mer analysis allows discovery of novel motifs and is not restricted by any database entry. Because most TF have core motifs between 5 and 8 bp, we use 6-, 7-, and 8 k-mer analysis to find enrichment of motifs of 8 or less base pairs. K-mers with significant variability were assembled and searched against multiple motif databases by Tomtom[41] for either similarity to known motifs or definition as a de novo motif.

## Data availability
Raw and processed data were available at NCBI Gene Expression Omnibus, accession number: GSE112091.

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

## Acknowledgements

This work was supported by NIH P50-HG007735 (to H.Y.C. and W.J.G.), R35-CA209919 (H.Y.C.), Parker Institute for Cancer Immunotherapy (to H.Y.C.), Swedish Research Council International Postdoctoral Fellowship and Starting grant (VR-2016-06794, VR-2017-02074 to X.C.), and King Abdulaziz University (H.C. and H.Y.C). We thank S. Kim (Stanford) for FACS access. The sequencing data were generated on an Illumina HiSeq 4000 that was supported by NIH award S10OD018220. H.Y.C. is an Investigator of the Howard Hughes Medical Institute.

## Author contributions

X.C, U.M.L, W.J.G. and H.Y.C conceived and designed the study. X.C, U.M.L. and E.L.L performed experiments. Y.W designed statistical analysis and wrote the scripts. X.C, U.M.L. and Y.W performed data preprocess. Y.W, X.C. and U.M.L performed data analysis with the scripts. A.N.S performed data analysis for Supplementary Figure 2f, l. E.L.L. H.C. and A.J.G. assisted with hypoxia experiments. X.C, U.M.L, Y.W. and H.Y.C wrote the manuscript with input from all authors. H.Y.C. supervised all aspects of this work.

## Additional information

**Competing interests:** The authors declare the following competing interests: H.Y.C. is an advisor to 10X Genomics and Spring Discovery. W.J.G. is an advisor to 10X Genomics. Stanford University hold a patent on ATAC-seq on which H.Y.C. and W.J.G. are named as inventors. H.Y.C. is a co-founder and member of scientific advisory board of Accent Therapeutics. The remaining authors declare no competing interests.

