## [Peer Review File · Nature Communications]

REVIEWERS' COMMENTS:

Reviewer #1 (Remarks to the Author):

The authors have adequately addressed my concerns.

Reviewer #2 (Remarks to the Author):

Reviewer 2 Comments to Authors

I was very supportive of this piece of work based on the fact that this is something I would like to use in my own research and it was a well-performed piece of technical research and a generally well-written manuscript. My comments were meant to improve the manuscript and the authors have done so to my satisfaction.

Reviewer 3 Comments to Authors,

Reviewer 3 from their comments also seems generally supportive, with some suggestions for presentation and clarification and the authors have addressed these sensibly.

The main scientific critique does not actually concern the methodology itself but rather the interpretation of the cellular composition of the cells in their breast tumor model. In point 5 the reviewer is suggesting that the variability in the HIF1 positive population could be being contributed to by an apoptotic population in this fraction, which is plausible. To address this the authors have performed a triple stain with P53, HIF1 and DAPI, allowing the cells to be further stratified as suggested by the reviewer and they show that the p53 signal they see does not represent an apoptotic fraction. In my view this addresses this point satisfactorily.

The other comments from reviewer 3 appear very helpful and straightforward and the authors have implemented them satisfactorily.